# HISTALIGN: Improving Context Dependency in Language Generation by Aligning with History

**David Wan**  **Shiyue Zhang**  **Mohit Bansal**

UNC Chapel Hill

{davidwan, shiyue, mbansal}@cs.unc.edu

## Abstract

Language models (LMs) can generate hallucinations and incoherent outputs, which highlights their weak *context dependency*. Cache-LMs, which augment LMs with a memory of recent history, can increase context dependency and have shown remarkable performance in diverse language generation tasks. However, we find that even with training, the performance gain stemming from the cache component of current cache-LMs is suboptimal due to the misalignment between the current hidden states and those stored in the memory. In this work, we present HISTALIGN, a new training approach to ensure good cache alignment such that the model receives useful signals from the history. We first prove our concept on a simple and synthetic task where the memory is essential for correct predictions, and we show that the cache component of HISTALIGN is better aligned and improves overall performance. Next, we evaluate HISTALIGN on diverse downstream language generation tasks, including prompt continuation, abstractive summarization, and data-to-text. We demonstrate that HISTALIGN improves text coherence and faithfulness in open-ended and conditional generation settings, respectively. HISTALIGN is also generalizable across different model families, showcasing its strength in improving context dependency of LMs in diverse scenarios.[1]

## 1 Introduction

Language modeling (LM), or language generation, requires decent context dependency. For both open-ended and conditional generation tasks, we want the model generation to be consistent with its previous generation or the input context. However, incoherence and hallucination problems are pervasive in current model generations (Holtzman et al.,

---

[1]Our code is publicly available at https://github.com/meetdavidwan/histalign

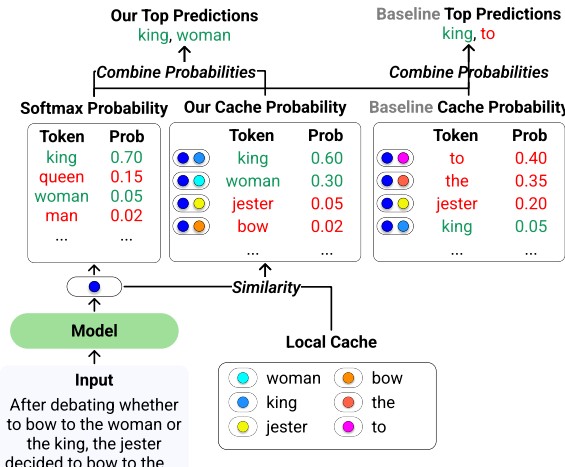

Figure 1: An illustration of HISTALIGN and baseline cache-LM. The input example is from Chang and McCallum (2022). Our HISTALIGN is able to assign high probabilities to both *king* and *woman*, and thus is able to tune down the weight of the hallucinated token *queen* from the softmax probability. Current cache language models (baseline) give high probabilities to irrelevant tokens in the cache and thus are at risk of producing hallucinated or incoherent tokens.

2020; Cao et al., 2018; Maynez et al., 2020), which suggests the weak context dependency of LMs.

Cache language model (Grave et al., 2017b, Cache-LM) is a simple yet effective method to improve context dependency by equipping LM with an additional memory of recent history (local context) and enabling it to directly "copy" from the history. Such models showed considerable improvement in language modeling and downstream generation tasks (Merity et al., 2017; See et al., 2017). However, since the introduction of Transformers (Vaswani et al., 2017), local memory has been less used due to the powerful self-attention mechanism, and more works have been focusing on leveraging long-term or external memory (Khandelwal et al., 2020; Yogatama et al., 2021). Nonethe-

less, Zhong et al. (2022) showed that using local memory on top of a Transformer is still beneficial.

In this paper, we focus on applying local cache to Transformer-based LMs and show that better alignment of the cache component leads to stronger gains. First, we show that cache-LM theoretically breaks the softmax bottleneck (Yang et al., 2018) that limits the capacity of any parametric LM to model highly context-dependent natural language. Then, we find that, in current cache-LMs, the signals provided by the memory component are minor, even when using the cache component during training (Zhong et al., 2022). We hypothesize that the main bottleneck comes from the misalignment of the current hidden states and those in the memory, because of which more relevant memories are not given higher weights than less relevant ones. We demonstrate this problem through a synthetic task: Ambiguous Template (Chang and McCallum, 2022), an example of which is shown in Figure 1. When asking the model to predict the next word given the context "*After debating whether to bow to the woman or the king, the jester decided to bow to the __* ," current cache-LM does not give the highest probabilities to the desired words *king* and *woman*. Instead, we find that irrelevant words, such as *to* and *jester* have high cache probabilities. When combining these probabilities with the original softmax, the desired words cannot be ranked as top tokens. We find that this problem exists in pre-trained LMs of various sizes, fine-tuned models, as well as models with cache augmented.

Next, we address this misalignment issue by proposing a new fine-tuning scheme, HISTALIGN, in which we augment the LM training objective with a contrastive loss to encourage the model to align the current hidden states with those in the history. As shown in Figure 1, our cache component gives higher probabilities for *king* and *woman* than other less relevant words in the cache. Unlike the typical contrastive loss that treats all negative examples equally, we propose to learn a ranking of negative tokens, i.e., more semantically similar tokens are ranked higher. As shown in Figure 2, when we align the space for the token *housing*, we want words such as *accommodations* to be closer than less relevant words like *children*. Hence, the cache can also be useful even when the exact target word is not present in the history. We demonstrate the stronger cache performance of HISTALIGN through the synthetic ambiguous template task and

showcase its strength in improving coherence for open-ended prompt continuation and faithfulness for abstractive summarization and data-to-text.

To summarize, our contributions are as follows:

- We discuss why cache-LM with local memory can improve context dependency through a softmax bottleneck lens.

- We show the misalignment problem present in current cache language models and their training strategy.

- We propose a new training method, HISTAL-IGN, based on order-informed contrastive learning, which alleviates the misalignment problem and makes better use of memories.

- We demonstrate that HISTALIGN improves the coherence of open-ended generation as well as the faithfulness of conditional generation, and it works across different model families and adds little computational overhead.

## 2 Related Work

**Cache-LM and Pointer Network.** Adding a cache component to a language model (LM) was first introduced for speech recognition (Kuhn and De Mori, 1990). Grave et al. (2017c) extended this idea to RNN-based neural LM, which they call *neural cache-LM*. Cache-LM predicts the next token by combining the RNN model's outputs with the similarities between the cache and the current hidden state. The cache saves tuples of hidden state and next token prediction, i.e., $(h_i, x_{i+1})$, from recent history (see Section 3.2). Essentially, the cache component enables the model to copy tokens from the history. Similar to cache-LM, a pointer network (Vinyals et al., 2015; Merity et al., 2017) also combines generating and copying of tokens but uses $h_i$ as a representation of $x_i$ (instead of $x_{i+1}$). This means that a pointer network requires learning additional transformations between the current representation and those in the past and a gating component for interpolation (Merity et al., 2017; See et al., 2017).[2] In contrast, cache-LM doesn't need extra parameters to be learned and can be applied directly at testing time. It is more efficient to

---

[2]Depending on the implementation, the model can have additional numbers of parameters that are quadratic to the number of hidden size for the projection matrix (for example, See et al. (2017) uses the concatenation of four hidden states for the gating module).

be used for larger cache sizes (i.e., extending cache-LM to long-term and external memory), and has been shown to perform better than pointer-network (Grave et al., 2017b; Zhong et al., 2022).

While cache-LM can be directly applied at test time, a recent work (Zhong et al., 2022) showed that it leads to more improvement when using cache during training time as well. Nonetheless, such proposed learning objectives for cache-LMs usually only provide distant supervision to the cache component. In contrast, we introduce direct supervision to the cache, which aligns the current representation with its history.

**LM with Local or External Memory.** Cache-LM and pointer network were originally proposed to only use hidden states from the local context, i.e., previous tokens in the input context. Though this technique has been proven to be helpful for language modeling and other language generation tasks (Gulcehre et al., 2016; Grave et al., 2017c; Merity et al., 2017; See et al., 2017), it has been less used after the Transformer architecture became popular, because the self-attention mechanism can attend to any token in the input context. Therefore, many works (Grave et al., 2017a; Khandelwal et al., 2020; Yogatama et al., 2021; Zhong et al., 2022; Min et al., 2022) proposed to use long-term or external memory beyond local context by applying retrieval techniques. Though our work can be extended to the external cache setting, we focus only on incorporating local memory, and we show that local memory is still helpful on top of Transformer because it breaks the softmax bottleneck (Yang et al., 2018) of parametric language models. A concurrent work (Chang et al., 2023) also demonstrates how a pointer network breaks softmax bottleneck by examples and empirical results, while we discuss this in a more mathematical way in Section 4.1.

**Context Dependency in Language Generation.** Existing language generation models demonstrate weak context dependency. For open-ended generation tasks, Holtzman et al. (2020) pointed out that strong LMs can produce very incoherent text following an input prompt. This incoherence issue has also been long observed in the story generation literature (Rashkin et al., 2020; Alabdulkarim et al., 2021). For conditional generation tasks, for example, summarization, Cao et al. (2018); Maynez et al. (2020) showed that around 30% and 70%

model-generated summaries contain hallucinations for two popularly used summarization datasets, respectively. Similar unfaithfulness problems have also been seen in data-to-text generation (Chen et al., 2020a), machine translation (Weng et al., 2020), etc. Though many approaches have been introduced to alleviate incoherence (Li et al., 2022a) or unfaithfulness (Cao and Wang, 2021; Wan and Bansal, 2022), in this work, we explore a simple yet general cache-LM method to increase context dependency for diverse tasks. The concurrent work (Chang et al., 2023) uses pointer network type of architectures to improve next-word distribution and summarization factuality. They modify the softmax head by using additional context-dependent embeddings. In contrast, we simply apply the original cache-LM architecture and improve it with a novel training objective.

## 3 Preliminaries

### 3.1 Language Modeling

We focus on autoregressive language modeling (LM). Here, for simplicity, we assume that the LM is decoder-only, i.e., the context of the current step is the generated tokens of previous steps. We show that the same approach can easily be generalized to encoder-decoder models in Section 4.3. Given the context $c_t = x_1, ..., x_{t-1}$, the probability of next token $x_t = w$ is predicted by a softmax head:

$$P_{lm}(w|c_t) \propto \exp(h_t^\top e_w) \qquad (1)$$

where $e_w$ is the output embedding of token $w$ and $h_t$ is the output context vector (hidden state) from the model at the $t$-th step. The model is trained by minimizing the cross-entropy loss: $l_{xe} = -\sum_t \log P_{lm}(x_t|c_t)$.

### 3.2 Cache Language Models

Cache language models augment a memory component to language models. Following Grave et al. (2017c), we consider cache to be a list of tuples of context vector and target token, $(h_i, x_i)$. Assume we only consider the history of the local context, then the local memory of $t$-th step is written as:

$$\mathcal{M}_{\text{local}} = \{(h_i, x_i)\}_{1 \le i \le t-1} \qquad (2)$$

Then, the next-token prediction aggregates the logits from the softmax head and the similarities be-

tween $h_t$ and those saved in the memory:

$$P_{clm}(w|c_t) \propto \exp(h_t^\top e_w) + \sum_{(h_i,x_i)\in\mathcal{M}_{\text{local}}} \mathbb{1}_{\{x_i=w\}}\exp(\text{sim}(h_t, h_i)) \quad (3)$$

where $\text{sim}(\cdot, \cdot)$ can be an arbitrary similarity function. Here, we follow Zhong et al. (2022) and use the scaled dot product: $\text{sim}(h_1, h_2) = \frac{h_1 \cdot h_2}{\sqrt{d}}$, where $d$ is the hidden dimension size.

While Grave et al. (2017c) only incorporated cache during evaluation, TRIME (Zhong et al., 2022) showed that it brings more benefits when also incorporated during training, i.e., minimizing $l_{trime} = -\sum_t \log P_{clm}(x_t|c_t)$. Here, we also use cache in both training and evaluation, but we improve the training objective by introducing direct supervision on the cache (see Section 4.2).

## 4 Our Methodology

### 4.1 Breaking Softmax Bottleneck

We first want to connect using local memory with the softmax bottleneck problem (Yang et al., 2018) and show that Transformer's self-attention cannot break this bottleneck, while the local cache can.

Parametric autoregressive language models (Section 3.1), including Transformer-based LMs, use a softmax function operating on context vectors (or hidden states) $\mathbf{H} \in \mathbb{R}^{N \times d}$ and output embedding matrix $\mathbf{E} \in \mathbb{R}^{V \times d}$. $N$ is the number of contexts, assuming every token in the training set has a different context, then $N$ is the number of tokens in the training set. $V$ is the vocabulary size, and $d$ is the hidden dimension size. Then, the next token probabilities form a log-probability matrix $\mathbf{A} \in \mathbb{R}^{N \times V}$ ($A_{tw} = \log P(w|h_t)$). Ideally, since every context is unique, the rank of $\mathbf{A}$ should be as large as $V$ (assuming $V < N$). However, as $\mathbf{A}$ is roughly equivalent to $\mathbf{HE}^\top$, its rank is strictly upper bounded by hidden size $d$ (please refer to Yang et al. (2018) for the formal proof). This low-rank problem greatly limits the LM's capacity to model highly context-dependent natural language. This can be seen in Figure 1, where *queen* achieves higher probability than *woman*. The reason for LM's difficulty in such bimodal distribution, as explained in Chang and McCallum (2022), is that the four words *king*, *woman*, *man*, *queen* tend to form a parallelogram in the embedding space, and if the model's hidden state wishes to be close to the output embeddings of *king* and *woman*, it will also be close to those of *man* and *queen*.

To break this bottleneck, one simple solution is to increase $d$, as we see larger models usually have better performance. Another solution proposed by Yang et al. (2018) and extended by Kanai et al. (2018); Yang et al. (2019); Chang and McCallum (2022) is to use multiple softmax heads – mixture of softmax (MoS), e.g., $P(w|h_t) \propto \exp(h_t^{(1)\top} e_w) + \exp(h_t^{(2)\top} e_w) + \exp(h_t^{(3)\top} e_w)$. Each $h_t^{(k)}$ is a different context vector. However, adding softmax heads is fairly computationally expensive. Comparing MoS to Eq. 3, we can see that adding $\exp(h_t^\top e_w)$ and $\exp(\text{sim}(h_t, h_i))$ resembles MoS without adding extra softmax heads. Another way to understand this connection is that when using local memory, $\mathbf{A}$ is roughly equivalent to $\mathbf{HE}^\top + \mathbf{HH}_c^\top$, where $\mathbf{H}_c$ are the hidden states in the local context.[3] Assuming $\mathbf{E}_c = \mathbf{E} + \mathbf{H}_c$, $\mathbf{A}$ becomes $\mathbf{HE}_c$. Different from $\mathbf{E}$, $\mathbf{E}_c$ is no longer a static output embedding matrix of size $V \times d$ but a context-dependent embedding tensor of size $N \times V \times d$. Hence, the rank of $\mathbf{A}$ is no longer upper bounded by $d$. Note that this connection also holds for using long-term or external memories.

### 4.2 HISTALIGN

Cache-LM combines the original softmax probabilities with the cache probabilities by aggregating the similarity scores between the current hidden state and those in the cache. To use the cache module effectively, the similarity function $\text{sim}(\cdot, \cdot)$ plays an important role in Eq. 3. If the similarities between the current hidden state and less relevant memories are higher than more relevant ones, it would steer the model away from selecting the most useful information from the cache. By assigning a high probability to the correct local memories, e.g., those corresponding to *king* and *woman* in the example of Figure 1, we can ensure that when the probabilities are combined, they will be scored higher than irrelevant and hallucinated tokens. However, we find that even when directly maximizing $\log P_{clm}$ (Zhong et al., 2022), there is no guarantee that the current representations are well aligned with relevant information stored in the memory, as shown by the baseline probabilities in Figure 1 (see Section 6.1 for more details).

Hence, to deal with this misalignment, we pro-

---

[3]This is because the logits under the cache-LM setting are the sum of the original token logits $HE^\top$ (first term of Eq. 3) and the cache logits $HH_C^\top$ (the second term of Eq. 3), and property 2 of Yang et al. (2018) showed that the log-probability matrix and logit matrix have similar ranks.

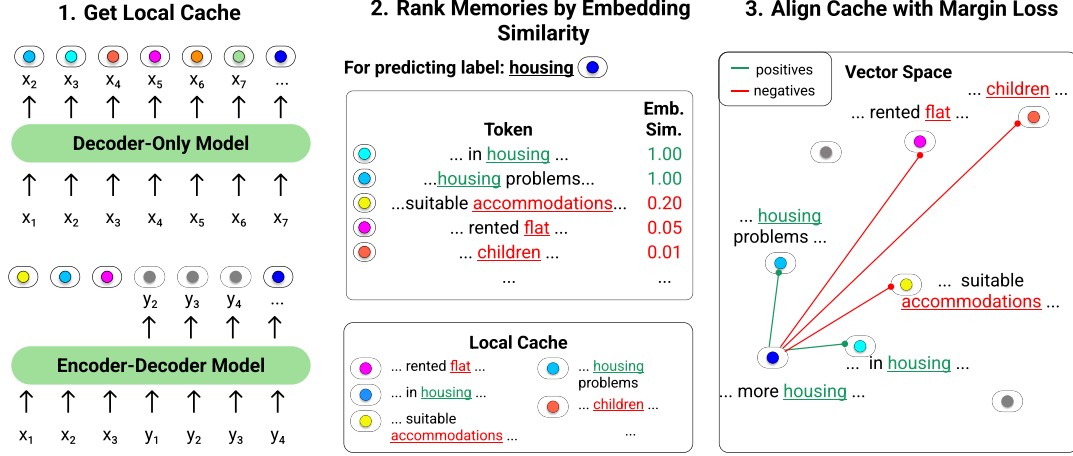

Figure 2: Illustration of our HISTALIGN training approach. We first get local cache by combining the hidden states in local context with their target tokens, and then rank them according to embedding similarity. The ranked memories are then used to train with the margin loss. This ensures that negative yet similar words (e.g. accommodations) will be closer in the vector space than irrelevant words (e.g. children).

pose a new contrastive objective that encourages higher similarities between the hidden states of similar target tokens. During training, given the current hidden state $h_t$ and the corresponding next token $x_t$, we construct a positive set $\mathcal{P}_t$ from caches by selecting memories with the same target token:

$$\mathcal{P}_t = \{(h_i, x_i)\}_{x_i=x_t, 1 \leq i \leq t-1} \quad (4)$$

All other memories are taken as negative examples. An example is shown in step 2 of Figure 2. For predicting the token *housing*, we have two previous mentions of the word housing, and the other words, including *flat, children, accommodations, etc.*, are considered as negative.

In the typical contrastive loss, such as InfoNCE (van den Oord et al., 2019), all negative examples are treated equally. However, we hope to learn an ordering of the negative examples – more similar examples are ranked higher than less similar ones. In the example in Figure 2, *accommodations* is more similar to *housing* than *children*. This ensures that even when predicting words that do not have previous mentions in the local cache, our model can still output a reasonable alternative.

To achieve this, we construct a ranking of memories by computing the cosine similarities between the embedding of the current target word and the embeddings of words in the cache, i.e., $\text{cosim}(e_t, e_i)$. After sorting tokens from the most similar w.r.t. semantic similarity to the least, we use

the following max-margin loss (Liu et al., 2022c):

$$l_{cont.} = \sum_t \sum_{i \in \mathcal{P}_t} \sum_{j>i, j \notin \mathcal{P}_t} \max\left(0, \text{sim}(h_t, h_j)\right.$$
$$\left. - \text{sim}(h_t, h_i) + \lambda_{i,j}\right)$$
(5)

where $\lambda_{i,j} = (j-i)\lambda$, and $\lambda$ is the margin tuned based on validation loss.

The final objective of HISTALIGN is a combination of the original LM cross-entropy loss $l_{xe}$ and this ranking-based contrastive loss:

$$l_{histalign} = l_{xe} + \alpha l_{cont.} \quad (6)$$

where $\alpha$ is a tunable weight of the contrastive loss. Note that during the inference time, we use Eq. 3.

### 4.3 Extension to Encoder-Decoder Models

HISTALIGN can be easily adapted to encoder-decoder models. For conditional generation tasks, the target text is usually short, hence, coherence is not a big issue. What is more crucial is whether the target generation stays true to the input context, e.g., the input document for summarization or the input table for data-to-text. Therefore, we define the local cache to be the input tokens and their corresponding encoder hidden states, as opposed to the output tokens and decoder hidden states for decoder-only models. We then calculate the similarity between the current decoder hidden state with those encoder hidden states stored in the cache.

## 5 Experimental Setup

Here, we describe the tasks and the experimental setups. Please refer to Appendix A for more details.

### 5.1 Tasks and Datasets

**Ambiguous Template** is a useful synthetic dataset collated by Chang and McCallum (2022), in which each example is generated using templates with diagonal words[4] from semantic analogy relations in the Google (English) analogy dataset (Mikolov et al., 2013). This is a simple yet effective setting to examine whether the model can copy the correct tokens from history and not hallucinate semantically similar tokens, e.g., *queen* and *man* of the example in Figure 1. Since the target words can always be found in the context, we can also evaluate the performance only with the cache component.

**Open-Ended Generation** evaluates the language modeling capability by asking the model to generate a continuation given a prompt (Holtzman et al., 2020; Su et al., 2022; Li et al., 2022b). We use WritingPrompts (Fan et al., 2018), and treat the first 50 tokens as the prompt and allow the model to generate up to 256 tokens using the canonical nucleus sampling ($p = 0.95$) (Holtzman et al., 2020).

**Abstractive Summarization** is the task of providing an abridged version of the input document. One crucial problem is 'hallucination', where the generated summaries contain facts or entities that are wrong or not present in the document (Cao et al., 2018; Maynez et al., 2020). We evaluate on two widely-used English News summarization datasets, XSum (Narayan et al., 2018) and CNN/DM (Hermann et al., 2015).

**Data-to-Text** is the task of describing structured data, where faithfulness is extremely important, as humans do not tolerate any hallucinations in cases such as describing medical reports or financial statistics (Thomson and Reiter, 2020). We evaluate on LogicNLG (Chen et al., 2020a).

### 5.2 Systems

We use GPT2-small and GPT2-large (Radford et al., 2019) for ambiguous template and prompt continuation, and we use BART-large (Lewis et al., 2020) for both summarization and data-to-text. For all tasks, we choose to finetune pre-trained LMs. The

---

[4]This refers to words that lie on the diagonal of a parallelogram in the embedding space. For example, for the tuple (queen, king, woman, man), the diagonal words are (king, woman) and (queen, man).

first baseline we compare to is fine-tuning with the original cross-entropy loss ($l_{xe}$ in Section 3.1), which is named by the original model name in our result tables. Then, we also compare to the most recent cache-LM learning objective, TRIME (Zhong et al., 2022) ($l_{trime}$ in Section 3.2).

### 5.3 Evaluations

**Ambiguous Template.** As a proof-of-concept experiment, we evaluate under both a full setting, using the combined probability in Eq. 3, as well as a cache-only setting, only using the cache similarity scores to predict the next token. We evaluate the performance via the accuracy of having the two diagonal words within the top-$k$ predictions (Acc@$k$), where $k = \{2, 5, 10, 25\}$. Ideally, we want to see 100% accuracy with $k = 2$, which indicates that the two diagonal words are the top 2 choices. Note that when only using the cache, a $k$ value of 50 would achieve perfect accuracy, as it would include the entire local history. In addition, we want to empirically verify that cache LM with local memory can break the softmax bottleneck. To this end, we calculate the rank of log-probability matrix $\mathbf{A} \in \mathbb{R}^{N \times V}$ (Section 4.1) using 500 examples (concretely, $N = 4750$ and $V = 50257$ for GPT-2 based models) under the full setting.

**Open-Ended Generation.** We mainly evaluate the *coherence* of model-generated continuations. Following Su et al. (2022), coherence is approximated by the cosine similarity of the SimCSE (Gao et al., 2021) sentence embeddings of the prompt and the continuation. In addition, following previous works, we report $n$-gram diversity (Meister et al., 2022) and MAUVE (Pillutla et al., 2021) scores for a more general evaluation. We hope HISTALIGN not to harm diversity and MAUVE. We also run **human evaluation** on Amazon MTurk to ask workers to compare the continuations generated by TRIME and HISTALIGN. More details can be found in Appendix B.1.

**Abstractive Summarization.** We mainly evaluate the faithfulness of generated summaries by three widely-used automatic metrics: FactCC (Kryscinski et al., 2020) and DAE (Goyal and Durrett, 2021), which are entailment-based metric; and Entity Precision (Nan et al., 2021, $P_{ENT}$), which calculates the percentage of entities in the summary that are present in the document. We also report ROUGE-L (Lin, 2004) for general content selection evaluation. Similarly, we conduct **human**

| Model | Full | | | | Cache-Only | | | | Full |
| | Acc@2 | Acc@5 | Acc@10 | Acc@25 | Acc@2 | Acc@5 | Acc@10 | Acc@25 | Rank |
|---|---|---|---|---|---|---|---|---|---|
| GPT2-Small | 50.00 | 62.20 | 68.80 | 76.96 | 0.00 | 35.57 | 50.56 | 75.71 | 762 |
| TRIME | 46.43 | 56.41 | 76.47 | 97.96 | 0.00 | 39.89 | 66.51 | **100.00** | 836 |
| HISTALIGN | **63.47** | **72.26** | **89.71** | **100.00** | **58.62** | **70.62** | **79.59** | 94.31 | **854** |
| GPT2-Large | 75.43 | 84.76 | 87.93 | 91.26 | 0.05 | 37.83 | 73.88 | **100.00** | 1280 |
| TRIME | 77.40 | 91.10 | 94.56 | 97.17 | 0.11 | 22.32 | 84.89 | **100.00** | **1377** |
| HISTALIGN | **82.22** | **92.84** | **96.57** | **98.34** | **82.15** | **92.51** | **99.94** | **100.00** | **1377** |

Table 1: Results on Ambiguous Template. HISTALIGN achieves the best performance in both full and cache-only settings. We also empirically show that TRIME and HISTALIGN break the softmax bottleneck.

| Model | Acc@2 | Acc@5 | Acc@10 | Acc@25 |
|---|---|---|---|---|
| LLaMA2-7B | 0 | 0 | 0 | **100** |
| TRIME | 0 | 0 | 0 | **100** |
| HISTALIGN | **100** | **100** | **100** | **100** |

Table 2: Cache-Only results on Ambiguous Template with LLaMA2-7B model.

**evaluation**, where we ask crowd workers to judge whether each summary (of 100 randomly selected examples) is faithful and informative. Please refer to Appendix B.2 for more details.

**Data-to-Text.** We mainly evaluate the faithfulness of model generations by NLI-Acc and SP-Acc (Chen et al., 2020a) and two more recent metrics – TAPEX-Acc and TAPAS-Acc (Liu et al., 2022a). NLI-Acc is an entailment-based metric pre-trained on TabFact dataset (Chen et al., 2020b) using TaBERT (Yin et al., 2020), and SP-Acc first parses the sentence into a logical program and evaluates the execution accuracy. TAPEX-Acc and TAPAS-Acc are entailment-based metrics trained with TAPEX (Liu et al., 2022b) and TAPAS (Eisenschlos et al., 2020), respectively. Same as previous works (Chen et al., 2020a), we report BLEU (Papineni et al., 2002) for a surface-level evaluation.

## 6 Results

We verify the strength of HISTALIGN at aligning the cache component and thus improve the next-token prediction on ambiguous template in Section 6.1, coherence in open-ended prompt continuation in Section 6.2, and faithfulness in abstractive summarization and data-to-text in Section 6.3 and Section 6.4, respectively.

### 6.1 Importance of Cache on Ambiguous Template

We show the results of the Ambiguous Template in Table 1. First, it can be seen that the original GPT2 model has pretty bad performance in the cache-only setting, especially considering Acc@2. This is ex-

pected because the original model is fine-tuned using the cross-entropy loss without the cache component involved, and thus applying cache at test time may not be helpful. Second, though TRIME (Zhong et al., 2022) generally outperforms the original model in the full setting, its cache-only Acc@2 and Acc@5 are similar to the original model. Considering that all target words are present in the history, this result indicates that despite the fact that TRIME uses cache during training, its cache component is still misaligned and has limited contributions to the final performance.

In contrast, HISTALIGN achieves high Acc@2 with only the cache module, substantially outperforming the original model and TRIME on both model sizes, which demonstrates the effectiveness of our contrastive loss for aligning memories better. As a result, HISTALIGN outperforms both baselines across all $k$ in the full setting. And the improvement holds for both model sizes, though with smaller gaps for the large model. This observation is consistent with our discussion in Section 4.1 that a larger model with a larger hidden dimension suffers less from the softmax bottleneck, while local memory can help break this bottleneck of any parametric LM. This is also empirically verified by the rank of the log-probability matrix reported in Table 1, where we see that the rank of the original model is upper-bounded by its hidden dimension (768 for GPT2-small and 1280 for GPT2-large), and having a local cache breaks this bottleneck. Finally, we present two qualitative examples in Table 9. See detailed discussions in Appendix C.

**Experiment on recent LLM.** We also fine-tune LLaMA2 7B model (Touvron et al., 2023). Interestingly, we find that LLaMA2 achieves 0% accuracy for Acc@{2,5,10} when evaluated zero-shot. After fine-tuning, the model achieves 100% accuracy without any cache. This is expected, as the task is a simple synthetic task, and the model, compared to GPT2-large, is 10x larger, and the hidden size

| Model | diversity | MAUVE | coherence |
|---|---|---|---|
| GPT2-small | $88.13_{\pm 0.12}$ | $86.62_{\pm 1.10}$ | $53.77_{\pm 0.29}$ |
| TRIME | $88.53_{\pm 0.14}$ | $86.76_{\pm 0.58}$ | $57.58_{\pm 1.05}$ |
| HISTALIGN | $\mathbf{90.07}_{\pm 0.19}$ | $\mathbf{87.46}_{\pm 0.80}$ | $\mathbf{61.30}_{\pm 0.15}$ |
| GPT2-large | $88.82_{\pm 0.07}$ | $86.18_{\pm 0.94}$ | $52.39_{\pm 0.10}$ |
| TRIME | $\mathbf{90.70}_{\pm 0.08}$ | $\mathbf{87.27}_{\pm 0.85}$ | $53.11_{\pm 0.19}$ |
| HISTALIGN | $89.41_{\pm 0.08}$ | $86.83_{\pm 1.02}$ | $\mathbf{53.51}_{\pm 0.05}$ |

Table 3: Automatic evaluation results of open-ended generation. Numbers are 3-run averages $\pm$ the 95% confidence intervals.

is 3.2x larger ($1280 \rightarrow 4096$). Thus, as mentioned in Section 4.1, the model alleviates the softmax bottleneck due to its larger hidden size.

However, we still observe the two problems with LLaMA2. First, the problem of softmax bottleneck still exists, as the rank of its output log-probability matrix $\mathbf{A}$ is still upper-bounded by its hidden size of 4096, as we find that its empirical rank is 3332. This means that it is still theoretically less expressive than highly context-dependent natural language. Second, TRIME is still not able to make good use of the cache, i.e., misalignment still exists. As shown in the Table 2, TRIME achieves 0% accuracy for Acc@{2,5,10} under the cache-only setting, which shows that the issue of misalignment is even more apparent for larger language models: Since the token logits perform well enough, the model does not learn to use the cache anymore. Nevertheless, as shown in the table, our training objective can enforce the use of the local cache and achieve 100% accuracy, which is consistent with our findings from smaller models.

The presence of these two issues showcases that there is still room for improvement on LM's context dependency, as HISTALIGN outperforms TRIME in making good use of cache.

## 6.2 Coherence in Open-Ended Generation

The results of prompt continuation can be found in Table 3. Across both sizes of the model, we observe an improvement in coherence with TRIME and a larger improvement with HISTALIGN. The effect of HISTALIGN is especially prominent for the smaller model, where coherence increases by 7.5 points compared to the original model, and 3.7 points over TRIME. This validates our hypothesis that HISTALIGN can improve the coherence of LMs. When looking at MAUVE, HISTALIGN improves by 0.8 points and 0.7 points over GPT2 and TRIME respectively when using small models. On the large model, while TRIME achieves the best

| Fluency | | | Coherence | | |
|---|---|---|---|---|---|
| Win↑ | Tie | Lose↓ | Win↑ | Tie | Lose↓ |
| 46.33 | 36.33 | 17.33 | 48.33 | 32.00 | 19.66 |

Table 4: Human evaluation results of open-ended generation. We conduct a pairwise comparison between HISTALIGN with TRIME ("Win" means humans prefer our HISTALIGN over TRIME) and show the percentage of passages that are judged as coherent and fluent. HISTALIGN is statistically significantly better ($p < 0.05$) than TRIME on fluency and coherence.

performance, HISTALIGN still improves over the original model by 0.7 points. A similar trend can be observed for diversity. Holistically, HISTALIGN improves coherence while maintaining similar diversity and MAUVE.

Besides automatic evaluations, we also conduct a **human evaluation**, the results of which are shown in Table 4. On both fluency and coherence, human raters prefer the continuations by HISTALIGN more than that by TRIME. This confirms the observation from the automatic evaluations that HISTALIGN does improve especially on coherence.

## 6.3 Faithfulness in Abstractive Summarization

The summarization results are shown in Table 5. TRIME improves faithfulness over the baseline on XSum, but the improvement is not clear on CNN/DM. In contrast, our HISTALIGN method greatly improves over the baseline, especially on DAE and $P_{ent}$, which are specifically targeted towards hallucinations. Concretely, we improve FactCC by 0.91 points, DAE by 4.78 points, and $P_{ent}$ by 3 points on the XSum dataset. HISTALIGN improves the metrics on CNN/DM as well though to a smaller degree. This shows that allowing the model to pay specific attention to previous contexts in the input is helpful in reducing hallucinations.

We note that the ROUGE-L score for HISTALIGN is lower than the original model. This ROUGE-faithfulness tradeoff has been observed by many previous works (Chen et al., 2021; Kryscinski et al., 2020; Wan and Bansal, 2022; Wan et al., 2023), where the reference summary inherently contains hallucinations and thus does not overlap highly with the more faithful generated summaries.

To confirm this, we conduct a **human evaluation**. The results are shown in Table 6. HISTALIGN achieves the best faithfulness score, which is statistically significantly better than BART. This confirms our observation from automatic metric results

| Model | XSum | | | | CNN/DM | | | |
|---|---|---|---|---|---|---|---|---|
| | Rouge-L | FactCC | DAE↓ | $P_{ent}$ | Rouge-L | FactCC | DAE↓ | $P_{ent}$ |
| BART | 36.41 | 22.16 | 67.96 | 72.72 | **30.63** | 72.63 | 6.98 | 93.53 |
| TRIME | **36.50** | 22.94 | 66.34 | 74.25 | 30.60 | 72.65 | 7.08 | 93.39 |
| HISTALIGN | 35.45 | **23.07** | **63.18** | **75.71** | 29.96 | **74.93** | **5.73** | **93.80** |

Table 5: Performance on abstractive summarization tasks. HISTALIGN consistently improves faithfulness over the two baseline methods on both datasets.

| Model | Faithfulness | Informativeness |
|---|---|---|
| BART | 19.33 | 63.67 |
| TRIME | 20.00 | 66.33 |
| HISTALIGN | 26.33* | 65.33 |

Table 6: Human evaluation results on XSum. * indicates that it is statistically significantly better (p < 0.05) than BART. Krippendorff's $\alpha$s are 0.52 and 0.34 for faithfulness and informativeness, respectively.

in Table 5. Though there is a small drop in informativeness, the difference between the three methods has no statistical significance.[5] This shows that the drop in automated metrics such as ROUGE-L does not necessarily mean a decrease in informativeness.

### 6.4 Faithfulness in Data-to-Text Generation

The results on LogicNLG are shown in Table 7. Similar to abstractive summarization, HISTALIGN can improve faithfulness on LogicNLG. Out of the four faithfulness metrics, HISTALIGN achieves the highest NLI-Acc, TAPEX-Acc, and TAPAS-Acc: HISTALIGN achieves 0.6 and 0.8 point improvements on TAPEX-Acc over BART and TRIME respectively, and a 1.74 point improvement on TAPAS-Acc over the BART model. In the meantime, HISTALIGN obtains the best BLEU scores.

### 7 Discussion and Conclusion

In this work, we improve the context dependency of LMs by introducing a novel cache-LM training objective, HISTALIGN, which improves the existing cache-LM objective by adding an order-informed contrastive loss for the cache component. On a synthetic dataset, we show that HISTALIGN is effective at retrieving the desired memories from the cache and breaking the softmax bottleneck. Furthermore, we demonstrate the effectiveness of HISTALIGN at improving the coherence of open-ended generation and improving faithfulness of abstractive summarization and data-to-text generation.

We want to emphasize a couple of salient points with the recent trend of pushing for larger and

---

[5]We use boostrap test (Efron and Tibshirani, 1993) to determine statistical significance in our paper.

| Model | BLEU-(1/2/3) | NA | SA | TX | TS |
|---|---|---|---|---|---|
| BART | 56.27/37.07/25.63 | 85.46 | **53.45** | 63.97 | 63.74 |
| TRIME | 56.12/36.84/25.29 | 84.55 | 52.85 | 63.74 | 65.11 |
| HISTALIGN | **56.65/37.56/26.25** | **85.67** | 53.12 | **64.58** | **65.48** |

Table 7: Performance on LogicNLG (data-to-text generation) evaluated by BLEU scores, NLI-Acc (NA), SP-Acc (SA), TAPEX-Acc (TA), and TAPAS-Acc (TS). HISTALIGN improves over two baselines on BLEU and three faithfulness metrics: NA, TX, and TS.

more powerful models. Firstly, attention mechanisms alone cannot break the softmax bottleneck, as shown in Table 2. Secondly, while increasing the model size can mitigate this bottleneck, the problem will persist unless we reach a size that truly encapsulates the complexity of human language. Cache-LM is a light alternative for breaking softmax bottleneck theoretically and improving context dependency empirically.

### Acknowledgments

We thank the reviewers and Haw-Shiuan Chang for helping with providing the Ambiguous Template data. This work was supported by NSF-CAREER Award 1846185, NSF-AI Engage Institute DRL-2112635, DARPA Machine Commonsense (MCS) Grant N66001-19-2-4031, and a Bloomberg Data Science Ph.D. Fellowship. The views contained in this article are those of the authors and not of the funding agency.

### Limitations

While we focus on the local memory to show that current LMs still benefit from better local context dependency, our method is also compatible with external memories, which can potentially further improve the performance of HISTALIGN in future work. We evaluate HISTALIGN using GPT2 and BART that at most consist of 774M parameters, which is smaller than the latest large LMs that can have billions of parameters. On the Ambiguous Template task, we do show that this problem exists for recent LLMs with LLaMA2 7B models and our method improves the cache alignment, but we

hope that in the future we can explore scaling up the approach on large LMs to various tasks. We believe that our method is still helpful for larger models. But as larger models suffer less from softmax bottleneck (Section 4.1), how much it can help is an interesting problem to study in the future. Another current limitation of this work is that due to the additional hyper-parameters (the $\lambda$ of the margin and the weight $\alpha$ of the contrastive loss), it becomes less straightforward to incorporate our HISTALIGN objective into pre-training compared to TRIME. The training objective also considers that each token has a fixed margin (and thus assumes that each token is equally different), which can be improved by dynamically adjusting the margins. Although fine-tuning is cheaper and we show effective gains using HISTALIGN in fine-tuning, how to use HISTALIGN to pre-train LMs is also an interesting future work direction.

## Ethical Considerations

As the OpenAI team pointed out, GPT-2 does not distinguish fact from fiction, so it can not support use cases that require the generated text to be true. In addition, GPT-2 reflects the biases inherent to the data they were trained on, so it can not be deployed unless the deployers first carry out a study of biases relevant to the intended use case. Though our HISTALIGN improves the coherence of GPT-2 generations, the above statement still holds. Similarly, despite that HISTALIGN improved the faithfulness of BART-large generations for abstractive summarization and data-to-text generation, such systems cannot be directly deployed and used in factuality-sensitive scenarios without further checks in place.

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

## A  Experimental Setup Details

Unless specified, we use Huggingface's Transformers library (Wolf et al., 2020) to train the models. We use the trainer's default setting, including AdamW optimizer (Loshchilov and Hutter, 2019) and a linear rate scheduler. We use mixed precision and deepspeed. We use RTX A6000 GPUs with 48GB memory and A100 GPUs with 80GB memory.

For hyperparameter tuning, we try learning rate of {1e-5,3e-5,5e-5} and $\lambda$ between {0.001,0.0001,0.00001}, and contrastive weight {0.5,1.0} for all tasks. For HISTALIGN, we use $\lambda$ of 0.001 and the contrastive weight $\alpha = 1$, unless otherwise specified.

### A.1  Ambiguous Template

The dataset consists of 122k, 250k, and 122k examples for train, dev, and test sets, respectively. The test set has no overlap of diagonal words with the training set. Following Chang and McCallum (2022), we freeze output vocab to prevent overfitting, and get loss only from the last token (the target token). We select the model based on validation loss. For GPT2-large-based models, training the original model took around 30 minutes, TRIME and HISTALIGN took around an hour with 4 RTX A6000. On GPT2-small-based models, training took 10 minutes, 15 minutes, and 15 minutes, for orig, TRIME, and HISTALIGN, respectively.

| Task | Dataset | Learning rate | Steps/Epochs | Warmup ratio | Batch size | Max input tokens | Max output tokens |
|---|---|---|---|---|---|---|---|
| Ambiguous Template | | 1e-5 | 5 ep. | 0.0 | 256 | - | 1024 |
| Open-ended Generation | WritingPrompt | 5e-5 | 3 ep. | 0.2 | 512 | - | 1024 |
| Summarization | XSum | 5e-5 | 15,000 steps | 0.0 | 128 | 512 | 64 |
| | CNN/DM | 3e-5 | 20,000 steps | 0.0 | 128 | 512 | 128 |
| Data-to-text | LogicNLG | 5e-5 | 10 eps | 0.0 | 64 | 500 | 200 |

Table 8: Hyper-parameters for all tasks.

| | Context | *Houston* and *Pennsylvania* are my favorites, and I especially love | The *brother* and the *granddaughter* are my favorites, and I especially love the |
|---|---|---|---|
| Full | GPT2-large | *Pennsylvania*, Harris, Pittsburgh | *brother*, niece, *granddaughter* |
| | TRIME | *Pennsylvania*, Philadelphia, Pittsburgh | *brother*, niece, *granddaughter* |
| | HISTALIGN | *Pennsylvania*, *Houston*, Harris | *brother*, *granddaughter*, niece |
| Cache-only | GPT2-large | and, I, *Houston* | the, *granddaughter*, I |
| | TRIME | I, and, *Houston* | the, *granddaughter*, favorites |
| | HISTALIGN | *Pennsylvania*, *Houston*, and | *granddaughter*, *brother*, the |

Table 9: Qualitative examples from Ambiguous Template. For both the full and cache-only settings, HISTALIGN retrieves the correct two tokens from the context as the top predictions.

## A.2 Prompt Continuation

WritingPrompts[6] (Fan et al., 2018) contain 273k, 16k, and 15k examples in the train, dev, and test sets. We use the full train and dev sets, while we sample 5000 examples from the test set for final evaluation to save time. We first train the models using the different objectives on the training set. We split the text into blocks of 512 tokens. For generation, we decode with nucleus sampling with $p = 0.95$ and three random seeds=\{0,1,42\}, and average the scores. Training the original small model takes around 1.5 hours, TRIME takes around 2 hours, and HISTALIGN takes around 3 hours. Training GPT-2 large, TRIME and HISTALIGN takes around 6 hours, 7 hours, and 11.5 hours on 2 A100s, respectively.

## A.3 Summarization

XSum is a news summarization dataset consisting of BBC articles and contains 204k/11k/11k examples in the train/dev/test set. CNN/DM consists of Dailymail and CNN articles and the dataset consists of 287k/13k/11k examples in the train/dev/test set. We use the official packages for the faithfulness metrics.[7] We calculate $P_{ent}$ by using spacy to extract entities and only consider [PERSON, FAC, GPE, ORG, NORP, LOC, EVENT] as the allowed entity types. We use Huggingface's Dataset library (Lhoest et al., 2021) for loading the XSum (Narayan et al., 2018) and CNN/DM (Hermann et al., 2015) datasets. And we use Huggingface's Metrics library for calculating ROUGE scores. Training the original model, TRIME, and HISTALIGN all took around 5 hours for XSum and training orig, TRIME and HISTALIGN all took around 4 hours for CNN/DM on 4 A6000s.

## A.4 Data-to-text

We follow Liu et al. (2022a) for pre-processing dataset, such as adding numerical pre-computation to the tables. We use a contrastive weight $\alpha = 0.5$. LogicNLG (Chen et al., 2020a) consists of 28k training, 4k validation, and 4k test examples. We use original evaluation scripts for the faithfulness metrics, and the BLEU calculation script provided by the original dataset.[8] Training the original model and TRIME took 2 hours, and HISTALIGN took around 5 hours on 2 A100.

## B Human Evaluation Details

For both human evaluations, we use Amazon Mechanical Turk to do the annotation. We have the same set of requirements: The workers need to be from the United States, have more than 10,000 number of HITS approved, and an approval rate greater than 98%.

[6] https://github.com/urvashik/knnmt/blob/master/examples/stories/README.md
[7] FactCC: https://github.com/salesforce/factCC. DAE: https://github.com/tagoyal/factuality-datasets.

[8] https://github.com/wenhuchen/LogicNLG/blob/master/evaluate.py

### B.1 Open-ended Generation

We use Amazon Mechanical Turk to annotate whether human prefers the continuation by TRIME or by HISTALIGN. We do not include the original model, since TRIME shows better performance on the automatic metrics. We select examples where the difference between their characters is less than 200 characters to ensure that the length is similar (since shorter texts will naturally be more coherent). We collect 3 annotations per example for 100 randomly selected examples, yielding 300 annotations. We take the percentage of passages that are judged as coherent and/or fluent.

We pay 0.5 USD per HIT, and the average time it takes is around 2.5 minutes, which yields an hourly rate of $\geq$ \$12 per hour. An example of the annotation page is shown in Figure 3.

### B.2 Summarization

We follow the same setup as Wan and Bansal (2022), and also use a qualification test where we rate the faithfulness of the selected generated summaries. Only workers with the correct annotation can perform the actual task.

We select the most important sentences and replace the less relevant sentences with an ellipsis to reduce the overload for the workers. We select ten most relevant sentences from the document by cosine similarity of the sentence embedding using SentenceTransformer[9] (Reimers and Gurevych, 2019) for each summary and then combine and show all the selected relevant sentences from each summary.

Each task consists of three unique workers, where we take the mean as the scores for this document. The final score is the mean factuality score across all documents. The average time for each task is around 2.5 minutes and we pay 0.5 USD per task, hence an hourly rate of $\geq$ \$12 per hour. An example of the annotation page is shown in Figure 4.

### C Qualitative Results on Ambiguous Template

We present two qualitative examples in Table 9. We see that both the original model and TRIME have difficulty in outputting the two correct words as the top two choices. This is also reflected by the cache-only results, where irrelevant words, such as *and*, *I*,

*the* get high probabilities. In fact, the cache similarities of the original model are similar to those of TRIME, again indicating that there is no guarantee of well-aligned memories, despite training with the cache. HISTALIGN nevertheless returns the two target words as the top two choices for both the full and cache-only settings, showing that the model benefits from the well-aligned memories through our contrastive objective.

### D Sample Outputs

We show sample outputs for prompt continuations in Figure 5, summarization in Figure 6 and Figure 7, and data-to-text in Figure 8.

---

[9]We use the all-mpnet-base-v2 model.

**Given the human-written prefix, please compare the systems and, if possible, choose which text is better. The system outputs are evaluated with respect to the following features: (1) Coherence; and (2) Fluency. In the following, we provide some guidelines regarding how to judge the quality of the system's result in terms of different features.**

**Coherence:** This metric measures whether the system's result is semantically and factually consistent with the human-written prefix. The system's result is perfectly in line with the semantic meaning defined by the prefix. And all its content is factually supported by or can be logically inferred from the prefix.

**Fluency:** This metric measures the fluency of the system's result. The system's result is human-like, grammatically correct, and very easy to understand.

Please note that:
1. It is NOT necessary that one paragraph is better than the other. You should choose "A and B are the same" if you think they are both very poor/good.
2. Paragraphs may contain facts that you are not familiar with. You do NOT need to verify those facts because **factuality is not a concern in this evaluation.**
3. Paragraphs are excerpts from longer text, so the ending part may not be complete. But **incompleteness should NOT affect your judgment.**

---

**Prompt:**
[WP] A crime fighting duo having a petty argument in the middle of fighting the villain. You are the villain. I watched in disbelief as the two vigilantes turned on eachother. "I'm getting the credit for this one!" Yelled

**A:**

one of the new vigilantes, with a smirk. My curiosity peaked when I saw them running towards me. "Just look at you! You're a badass!" My mind was spinning, but I kept looking. "You... you're my son!" This time it had to be a boy, he wasn't too big for that kind of thing! "What the... you'ren't a boy!" Both villains began to yell at each other, but I had my best friend, the one who just started laughing, still running with her hands. "Shut up, bitch! Just get the credit!" I yelled, and he turned to her, who just started running. She threw him a towel and ran after him. "What the... what the hell are you doin 'with that towel?", I shouted as I chased after him, but he was gone. I followed him until I came across a room with two masked men in it. One of the masked man was a big hairy black beast with a large, dark face, while the other was a small, tall black figure. "What the hell are you doin 'with that towel!?" I yelled, and they kicked at me, making my skin crawl. "What the hell are you doing

**B:**

the man in green of my world with white to his face. "The one who killed us had the same sentence!" I thought I had seen that one again. But the man in green did not. He was still a young girl and a child. He wore the same attire. As I watched that man make the deal, they said this is my fate. "I'll have my revenge tomorrow." I told him as he looked at my reflection. "I'm going to kill you when you die!" I whispered as I tried to put my finger on the knife. "You were too slow. You got too many things to be done. This is your revenge." The woman in purple came forward and I turned to my partner in white. "Look, I'll be back on the next day!" I said as I placed my hand on her right shoulder. I thought, I did not have to run. With that as my face was completely open, I threw my fist in my face, and I shot the man on the back with his hands and head. He was almost killed. I had taken a moment to think. "I've just killed all this people you've met on the playground. You've ruined those people. All you've done is kill them

**Coherence:**
○ **A is better**          ○ **B is better**                          ○ **A and B are the same**
**Fluency:**
○ **A is better**          ○ **B is better**                          ○ **A and B are the same**

Submit

Figure 3: Human annotation page for evaluating coherence and fluency for prompt continuation.

**Please evaluate whether the three summaries are coherent, consistent with the information in the article, and are informative.**

The article has been separated into sentences and displayed line-by-line. We have shortened the text for your convenience by removing the least related sentences, indicated by ellipsis (...).

Note that the five summaries shown are different and **not in any particular order**.

Please evaluate the summary in the following three categories:

**Coherence**
Irrespective of the given document, please select **coherent** if the summary is fluent. The summary has better structure and flow, and is easier to follow. The facts are presented in a more logical order.

**Consistency/Factuality**

Please **avoid using general knowledge**, and only consider in the context of the provided document.

Select **not consistent** if facts in the summary are not supported by the document, such as cases like these:

- The summary **contradicts** information in the document. The summary might say "A fire broke out in Seattle", but an document says it broke out in Portland. Or the summary might say "the Republicans won the election", but the document indicates the Democrats won instead.
- The summary **adds (hallucinates)** a fact that is not mentioned anywhere in the document. For example, the summary might say thet "A fire broke out at 2am", but the document doesn't mention the time when the fire broke out.

**Informativeness**

Please select **informative** if the summary expresses the main points of the document. Summary should contain relevant and important information and few unimportant details.

If you select the summary to be not consistent with the document, please only consider the consistent information when evaluating this category.

**Document:**
...
First-team coach Andy Smith, goalkeeping coach Marco Tabuas and fitness coach Maykel Moreira have now also left Vale Park, the League One club has confirmed.
The three all arrived when Ribeiro was appointed in the summer.
"We thank them for their time with us," a club statement said.
"They helped assist the club to their best home start to a season ever at Vale Park and progression to the FA Cup third round."
Vale face a home game with Chesterfield on Friday before a trip to Oldham on Monday, prior to playing Championship side Huddersfield Town in the FA Cup on 7 January, They have lost their last three league games to slip to 17th in the table - just six points above the relegation zone.
Ribeiro's former assistant, Michael Brown, has been placed in temporary charge of the Burslem-based club.

**Summary:**
Port Vale manager Bruno Ribeiro has left the club by mutual consent after less than two months in charge.

| Coherence: | Consistency: | Informativeness: |
|---|---|---|
| ○ **coherent** | ○ **consistent** | ○ **informative** |
| ○ **not coherent** | ○ **not consistent** | ○ **not informative** |

**Summary:**
Port Vale head coach Bruno Ribeiro has left the club by mutual consent after less than two months in charge.

| Coherence: | Consistency: | Informativeness: |
|---|---|---|
| ○ **coherent** | ○ **consistent** | ○ **informative** |
| ○ **not coherent** | ○ **not consistent** | ○ **not informative** |

**Summary:**
Port Vale have sacked manager Bruno Ribeiro after just three months in charge.

| Coherence: | Consistency: | Informativeness: |
|---|---|---|
| ○ **coherent** | ○ **consistent** | ○ **informative** |
| ○ **not coherent** | ○ **not consistent** | ○ **not informative** |

Figure 4: Human annotation page for evaluating faithfulness and informativeness for abstractive summarization.

**Prompt**: Every time you die, you are transported to the time of the first event that ultimately caused your death. One day, you are hit by a car and sent 12 years into the past. It's been a strange life; constantly setting

**GPT2-Large**: up and killing other people. But it's done, and now it ' s time to settle down in this new world. I ' d never actually thought about my past life before, I just knew that I went back and lived, but this time there ' s a complication. I ' ve been to many places in my life, so why is there such a big problem? It ' s the weirdest thing, I could go anywhere in time and never get the feeling, but this time it ' s weird. The cars are just like the last time, the ones that hit me, there ' s just a hole in the middle of them. I ' m in a small, abandoned office, and the walls aren ' t quite made out of cement. I think that ' s a nice touch ...

**TRIME**: up and killing other people. But it's done, and now it's time to put the final touches on this latest creation. I'll be frank, I don't really know what I'm supposed to do. A couple months ago, I was on my way to my parents, and there was an accident on the side of the road. It was some kind of accident; the car was hit by a drunk driver, and it fell on my head. Luckily, I was alive. I was given a few months to live, by the hospital. I can live with that, at least until I can figure out what happened. Anyway, I'm now waiting for the day when I meet my next death, and I have to be prepared for the worst. I can try to avoid death ...

**HISTALIGN**: up and stopping things, trying to prevent every mistake that might happen. If it weren't for the fact that I was always so careful with everything, I might be a normal person who just goes about their business. A normal person like you wouldn't have the ability to go back in time to prevent yourself from ever having died in the first place. Every mistake that might have happened could have killed me then and there, and it wasn't an easy road to travel down. I was lucky enough to have only been born on July 4th, 1996. I've been trying to do the right thing all these years. I've avoided death on more than one ...

**Prompt**: Crown Minotaur Expedition The first week of the expedition was slow going. The terrain was rocky and unforgiving, as if the Mother Nature had forgotten to smooth out this patch of land. The Minotaurs had the most trouble due to the

**GPT2-Large**: steep slopes of the mountain pass. When a Minotaurel came up the mountain pass, she would pass straight through, just passing through the pass. It was just one of many impassable ridges, that the Miatarel would face to traverse the pass. The only reason it was never impassable was due to the massive, hairy beasts, which covered half of the pass, and caused many people to leave the mountains. This, plus the numerous Minotaurel who passed through the pass would create an increasingly high attrition rate on the expedition. Still, the Miatarel had learned of the pass through the passing ...

**TRIME**: steep slopes of the mountain pass. When a mountain pass was at its highest, the Minotaurs were no longer able to pass into the valley below. It was just too steep. Luckily there was plenty of rock that they could use as a ramp and they would be able to crawl up with their arms, legs, and wings intact. The trip began on the third day after lunch. The trail was still rough, as many of the boulders were nearly a foot high. By now they could barely lift the weight of their bodies, much less their bags full of provisions and equipment. The road ahead of them was now covered in small puddles. This was typical of these ...

**HISTALIGN**: steep slopes of the mountain pass. When a mountain pass was at its highest, the Minotaurs were no longer able to pass into the valley below. It was just too steep. Luckily, they were able to set up camps and set out in the first few days. This was only going to be a matter of time until they realized where they were going. This place was far away from anything they knew of. The only light was the lanterns on their back, and the lanterns were only good for a few minutes. As for how they got there, no one is quite sure. They have not been able to find any of the equipment they carried when they left ...

Figure 5: Sample outputs for open-ended prompt continuation.

**Document**: David Lipton, second in command at the IMF, outlined some of these risks in a speech to the National Association for Business Economics in Washington on Tuesday. "The IMF's latest reading of the global economy shows once again a weakening baseline," he said. "We are clearly at a delicate juncture." The comments come after weaker-than-expected trade figures from China showing that exports plunged by a quarter from a year ago. The IMF has already said it is likely it will downgrade its current forecast of 3.4% for global growth when it next releases its economic predictions in April. The dismal picture is one that has on-going ramifications for businesses and industries that bet on China's growth story. Read more from Karishma: Why a story about bulk shipping matters

**BART**: The International Monetary Fund (IMF) has warned that the global economy is at a "critical juncture".
**TRIME**: The International Monetary Fund (IMF) has warned that the global economy is in a "dangerous situation".
**HISTALIGN**: The International Monetary Fund (IMF) has warned that the global economy is in "a delicate juncture".

**Document**: Coventry University's Scarborough campus has been built on the town's former Weaponness Park and Ride site. About 200 students have begun courses at the site, though it is expected to eventually be home to more than 2,000 students. The building, which includes engineering and science labs, a mock law court and a library, is part of a £50m sports and education facility. Professor Craig Gaskell said: "Launching our new state-of-the-art building is a huge milestone for us and demonstrates our commitment to Scarborough and the Yorkshire coast area." A new University Technical College has been built nearby and Scarborough Athletic FC's new 2,000-seater stadium is also under construction on the site. Coventry University also has a campus near London's Liverpool Street Station and recently announced it will open a campus in Dagenham in September 2017.

**BART**: A university has officially opened its first campus in North Yorkshire.
**TRIME**: A new university campus has been officially opened in North Yorkshire.
**HISTALIGN**: A university campus on the Yorkshire coast has opened to the public.

Figure 6: Sample outputs for XSum summarization.

**Document**: Bayern Munich will be without versatile left-back David Alaba for the closing stages of the season after he was ruled out for around seven weeks with a torn knee ligament. Alaba, out for three months earlier this season after a partial ligament tear in his right knee, was injured while playing for Austria in their 1-1 draw against Bosnia on Tuesday. 'I am bitterly disappointed,' the 22-year-old told the club's website. 'But I have a big aim and that is to be there at the season finale.' David Alaba is tackled by Emir Spahic's tackle during Austria's 1-1 draw against Bosnia-Herzegovina . Marc Janko (left) celebrates with Martin Harnik and Alaba after he scored against Liechtenstein last Friday . He is set to start rehabilitation work in about a month, Bayern said. Alaba had been in superb form for Bayern since his comeback, helping them stay firmly on track for a treble of titles with some spectacular goals. Bayern, who are top of the Bundesliga with eight games left and through to the German Cup last eight, face Porto in the Champions League quarter-finals later this month. Alaba (left) has been in excellent form for Bayern this season as the Bavarian giants chase a treble of titles .

**BART**: David Alaba has been ruled out for around seven weeks with a torn knee ligament. Alaba was injured while playing for Austria in their 1-1 draw against Bosnia on Tuesday. Bayern Munich face Porto in the Champions League quarter-finals later this month.

**TRIME**:David Alaba has been ruled out for around seven weeks with a torn knee ligament. Alaba was injured while playing for Austria in their 1-1 draw against Bosnia on Tuesday. Bayern Munich face Porto in the Champions League quarter-finals later this month. The 22-year-old is set to start rehabilitation work in about a month.

**HISTALIGN**: David Alaba has been ruled out for seven weeks with a torn knee ligament. Austria defender was injured in their 1-1 draw against Bosnia-Herzegovina. Bayern Munich face Porto in the Champions League quarter-finals later this month.

**Document**: Derry Mathews has won the interim WBA lightweight title after being handed a unanimous points decision over Tony Luis in Liverpool. Mathews saw Richar Abril twice pull out of a fight due to illness to be striped of the belt and just this week replacement Ismael Barroso was forced to pull out after failing to obtain a visa. Derry Mathews celebrates after victory over Tony Luis at Echo Arena on Saturday night . Mathews celebrates with his family after claiming the interim WBA lightweight title . Luis was drafted in at the last minute and while the Canadian will have won a lot of fans with his relentless style, he did not win over the judges and Mathews was handed a 114-112, 114-112, 115-112 decision. Mathews made the brighter start and kept Luis contained behind the jab while landing a decent uppercut at the end of the second round. Mathews throws a right punch during interim WBA lightweight title fight in Liverpool . Luis (right) was drafted in after Richar Abril and Ismael Barroso both pulled out . The Liverpudlian continued to still largely control the fight but Luis showed plenty of aggression himself, coming forward and making life difficult for Mathews, landing a good uppercut of his own at the end of the sixth. Luis' pressure and all-action style was beginning to tell on Mathews, who was more and more often going to the ropes and both men were deducted a point in the eighth, Mathews for a low blow and Luis for use of the head. Luis was in the ascendency in the closing rounds and looked to be pulling some points back on the cards but it was not enough as Mathews was handed a tight decision after a thrilling encounter.

**BART**: Derry Mathews beat Tony Luis in Liverpool on Saturday night. Mathews saw Richar Abril twice pull out of a fight due to illness to be striped of the belt. Ismael Barroso was forced to pull out after failing to obtain a visa.

**TRIME**: Derry Mathews beat Tony Luis in Liverpool on Saturday night. Mathews saw Richar Abril twice pull out of a fight due to illness. Ismael Barroso was forced to pull out after failing to obtain a visa. Luis was drafted in at the last minute but did not win over the judges.

**HISTALIGN**: Derry Mathews beat Tony Luis by unanimous points decision at Echo Arena. Mathews saw Richar Abril and Ismael Barroso pull out of the fight due to illness and illness. The Canadian was handed a 114-112,114-112 and 115-112 decision by the judges. Liverpudlian Luis was drafted in at the last minute after the pair pulled out.

Figure 7: Sample outputs for CNN/DM summarization.

| | home team score | away team | away team score |
|---|---|---|---|
| | 11.12 (78) | st kilda | 21.11 (137) |
| | 8.9 (57) | geelong | 8.8 (56) |
| Table for "1928 vfl season": | 11.15 (81) | richmond | 10.13 (73) |
| | 22.17 (149) | hawthorn | 11.13 (79) |
| | 18.18 (126) | fitzroy | 11.13 (79) |
| | 11.17 (83) | essendon | 18.11 (119) |

**Reference**: St Kilda had the highest Score as an Away Team in the 1928 Vfl Season

**BART**: Hawthorn had the lowest Away Team Score of any team in the 1928 Vfl Season
**TRIME**: Geelong had the lowest Score of 8.8 (56) while Hawthorn had the highest Score of 11.13 (79)
**HISTALIGN**: St Kilda was the Away Team with the highest Score in the 1928 Vfl Season

| | name | matches |
|---|---|---|
| | john hearne | 29 |
| | tom richardson | 23 |
| Table for "1893 english cricket season": | johny briggs | 28 |
| | arthur mold | 28 |
| | bill lockwood | 27 |

**Reference**: John Hearne, played in more Match than any other Player, with 20 9

**BART**: Bill Lockwood and Arthur Mold had the same number of Match
**TRIME**: Bill Lockwood and Arthur Mold both played 27 Match in the 1893 English Cricket Season
**HISTALIGN**: John Hearne had the most Match with 29

Figure 8: Sample outputs for data-to-text generation.