# OpenReview forum: "HistAlign: Improving Context Dependency in Language Generation by Aligning with History"
_EMNLP/2023/Conference — EMNLP 2023 Main_

### Official Review · Reviewer_GMge · 2023-07-29

**Soundness:** 4

**Excitement:**

4: Strong: This paper deepens the understanding of some phenomenon or lowers the barriers to an existing research direction.

**Paper Topic And Main Contributions:**

The paper presents an approach for improving coherency and reducing hallucinations in a text generated by Transformer language models (LMs). The approach is based on Cache-LM (Grave et al., 2017b), a method for equipping a RNN with an additional memory of recent history. Cache-LM combines the standard softmax probabilities with probabilities of copying a token from cache, using similarity of hidden states to determine the relevancy of cached token for the current context.

Here, the authors adapt Cache-LM for Transformer decoder and encoder-decoder architectures. On a variety of tasks (including a synthetic task for semantic analogy, open-ended generation, abstractive summarization, and data-to-text generation), the authors show that LMs (GPT-2 and BART) equipped with the caching mechanism achieve better factuality and coherency while retaining text diversity and lexical similarity to the reference.

**Questions For The Authors:**

A) What is an advantage of the Cache-LM compared to the pointer network, and have you considered including a model equipped with a pointer network as one of the baselines?

**Reasons To Accept:**

- The paper demonstrates that the proposed method robustly improves coherence of text generated by mid-sized Transformer models, both for open-ended and conditional generation.
- The method can be simply applied to both decoder-only and encoder-decoder architectures.
- The authors clearly explain the reasoning behind the design choices and arrive to reasonable conclusions on the basis of the performed experiments.
- The paper itself is coherent and easily readable.
- The code for the experiments is provided in the supplementary.

**Reasons To Reject:**

- I am missing a stronger distinction between the Cache-LM and the pointer network. As far as I understood from §2 (l. 138-142), the formulations are fairly similar. There is also a concurrent work using a pointer network for solving similar problems (l. 198-201). I thus believe that the authors should devote more space for explaining the differences between these two architectures. (I am willing to increase the soundness score if this point is made more clear.) [After rebuttal: I have increased the soundness score to 4.]
- As the authors point out, the gains from the proposed method are diminishing as LMs grow in size, since the problem can be also tackled by increasing the size of the context vector.

**Reproducibility:**

5: Could easily reproduce the results.

**Reviewer Confidence:**

3: Pretty sure, but there's a chance I missed something. Although I have a good feel for this area in general, I did not carefully check the paper's details, e.g., the math, experimental design, or novelty.

---

> ### Author Rebuttal · Authors · 2023-08-28
>
> We thank the reviewer for the helpful and detailed comments and for appreciating the robustness (on multiple NLG tasks) and flexibility (in adapting to different architectures) of our method! Please see below for responses to your questions:
>
> **Difference between pointer network and cache-lm**
> > I am missing a stronger distinction between the Cache-LM and the pointer network. As far as I understood from §2 (l. 138-142), the formulations are fairly similar. There is also a concurrent work using a pointer network for solving similar problems (l. 198-201). I thus believe that the authors should devote more space for explaining the differences between these two architectures. (I am willing to increase the soundness score if this point is made more clear.)
>
> > A) What is an advantage of the Cache-LM compared to the pointer network, and have you considered including a model equipped with a pointer network as one of the baselines?
>
> To expand on our points from Lines 138-142: As noted in prior studies such as Grave et al., 2017a and Zhong et al., 2022, pointer network uses the current hidden activation ($h_i$) as a representation of the current input ($x_i$), while cache-LM use it to represent the $x_{i+1}$. Hence, a pointer network requires additional learning of a transformation between the current representation and those in the past. Also, it usually requires a learned gating component for interpolation (Merity et al., 2017; See et al., 2017). Therefore, adding a pointer network to a model means adding quite a number of new parameters. Depending on the implementation, the model will have additional numbers of parameters that are quadratic to the number of hidden size for the projection matrix (for example, See et al., 2017 uses the concatenation of four hidden states for the gating module).
>
> In contrast, cache-LM doesn't need extra parameters to be learned and can be applied directly at testing time. It is more efficient to be used for larger cache sizes (i.e. extending cache-LM to long-term and external memory).
>
> Furthermore, previous works (Grave et al., 2017a, Zhong et al., 2022) have shown that the pointer network performs worse than cache-LM, and we thus mainly compare to cache-LM baselines.
>
> We will make this point clear in the camera-ready version.
>
>
> **Diminishing performance with larger model**
>
> > As the authors point out, the gains from the proposed method are diminishing as LMs grow in size, since the problem can be also tackled by increasing the size of the context vector.
>
> We agree that local cache will become less useful when the model becomes larger. That being said, we want to emphasize a couple of salient points. Firstly, as highlighted in Lines 250-278, attention mechanisms alone cannot break the softmax bottleneck. Secondly, while increasing the model size can mitigate this bottleneck, the problem will persist unless we reach a size that truly encapsulates the complexity of human language (lines 279-281). Hence, we think that as the model becomes larger, its context dependency gets better but there will still be room for improvement. Cache LM is a light alternative for breaking softmax bottleneck theoretically and improving context dependency empirically. How to make it stay useful for larger models is an interesting future research direction. Besides, smaller (e.g., distilled) models by far remain to be useful in real-world applications. We hope our method can be helpful in these cases.
>
> We will make this point clear in the camera-ready version.

---

### Official Review · Reviewer_gaqX · 2023-08-04

**Soundness:** 4

**Excitement:**

4: Strong: This paper deepens the understanding of some phenomenon or lowers the barriers to an existing research direction.

**Paper Topic And Main Contributions:**

This paper addresses the problem of incoherence and hallucination in state-of-the-art language models (LM). Since the language model is trained on huge text, it generates the next word based on the entire training memory rather than giving the relevant word within the current context. State-of-the-art uses cached language model to address this issue. On top of this, this paper proposes a discriminative training criteria (with a max-margin loss that adjusts the probability score of not only the correct token but also similar tokens ranked according to similarity) to further improve the performance.

**Reasons To Accept:**

a1. The methodology is somewhat novel; the rationale is clear and reasonable.
a2. The author has evaluated LM performance on 4 different LM-based tasks, and achieve significant improvement in all of them.
a3. See b1, this technique might be particularly effective on smaller models or dataset.

**Reasons To Reject:**

b1. In LLM-based models such as ChatGPT, attention is effectively doing the caching operation (by attending to relevant tokens in the context, it generates the next word less according to the whole training memory) as well as contrastive training (the weight updates on less attended tokens are smaller). Therefore, I am afraid when both the LM and the training data are sufficiently large, the benefit of cached LM and contrastive training might diminish.
b2. In Equation (5), the definition of lambda_{i,j} is a bit ad-hoc. It assumes nth-ranked tokens are equally spaced in the probability space which might not be the case.

**Reproducibility:**

4: Could mostly reproduce the results, but there may be some variation because of sample variance or minor variations in their interpretation of the protocol or method.

**Reviewer Confidence:**

3: Pretty sure, but there's a chance I missed something. Although I have a good feel for this area in general, I did not carefully check the paper's details, e.g., the math, experimental design, or novelty.

---

> ### Author Rebuttal · Authors · 2023-08-28
>
> We thank the reviewer for the helpful and detailed comments and for appreciating the soundness of our method as well as the strong experiment results! Please see below for responses to your questions:
>
> **Diminishing performance with larger model**
>
> > b1. In LLM-based models such as ChatGPT, attention is effectively doing the caching operation (by attending to relevant tokens in the context, it generates the next word less according to the whole training memory) as well as contrastive training (the weight updates on less attended tokens are smaller). Therefore, I am afraid when both the LM and the training data are sufficiently large, the benefit of cached LM and contrastive training might diminish.
>
> We agree that local cache will become less useful when the model becomes larger. That being said, we want to emphasize a couple of salient points. Firstly, as highlighted in Lines 250-278, attention mechanisms alone cannot break the softmax bottleneck. Secondly, while increasing the model size can mitigate this bottleneck, the problem will persist unless we reach a size that truly encapsulates the complexity of human language (lines 279-281). Hence, we think that as the model becomes larger, its context dependency gets better but there will still be room for improvement. Cache LM is a light alternative for breaking softmax bottleneck theoretically and improving context dependency empirically. How to make it stay useful for larger models is an interesting future research direction. Besides, smaller (e.g., distilled) models by far remain to be useful in real-world applications. We hope our method can be helpful in these cases.
>
> We will make this point clear in the camera-ready version.
>
>
> **Training objective**
>
> > b2. In Equation (5), the definition of $\lambda_{i,j}$ is a bit ad-hoc. It assumes nth-ranked tokens are equally spaced in the probability space which might not be the case.
>
> We agree with the reviewer regarding the potential enhancement of the training objective. Our primary rationale for adopting this training objective from Liu et al., 2022c centers on its emphasis on the ranking of tokens, rather than the margins between tokens.

---

### Official Review · Reviewer_DVaq · 2023-08-04

**Soundness:** 4

**Excitement:**

4: Strong: This paper deepens the understanding of some phenomenon or lowers the barriers to an existing research direction.

**Paper Topic And Main Contributions:**

This paper proposes to improve the context dependency of cache LMs. The authors connect the cache LM approach and the softmax bottleneck in the Transformer architecture. The main technical contribution is a contrastive learning objective encouraging higher similarity between histories corresponding to the same output token. Empirical results show a significant improvement in retrieving suitable memory and in downstream task coherence and faithfulness.

**Questions For The Authors:**

A) Line 291-310, can you add some more details on why $A$ is roughly equivalent to $HE^T + HE_c^T$ and what $E_c$ (hidden states in the local context) means here?

**Reasons To Accept:**

- In general, the paper is well-motivated. With a clear analysis, the paper proceeds to identify the root cause of the lack of context dependency and proposes the contrastive loss proposed is novel and logically sound. The design is simple but effective, bringing inspiration that the community can benefit from.
- The experiment results are strong, indicating the efficacy of the methodology.
- The paper is well written. The logic and design details are presented in a way that is easy to follow.

**Reasons To Reject:**

- The baselines are limited to GPT2 and BART. It would be interesting to see the performance of more recent works of LLM such as LLaMA.

**Reproducibility:**

4: Could mostly reproduce the results, but there may be some variation because of sample variance or minor variations in their interpretation of the protocol or method.

**Reviewer Confidence:**

4: Quite sure. I tried to check the important points carefully. It's unlikely, though conceivable, that I missed something that should affect my ratings.

---

> ### Author Rebuttal · Authors · 2023-08-28
>
> We thank the reviewer for the helpful and detailed comments and for appreciating the novelty and efficacy of our method as well as the strong experiment results! Please see below for responses to your questions:
>
> **Clarification on breaking the softmax bottleneck**
>
> > A) Line 291-310, can you add some more details on why $A$ is roughly equivalent to $HE^T + HE_{c}^{T}$ and what $E_c$(hidden states in the local context) means here?
>
> We would like to clarify that $A$ is roughly equivalent to $HE^T + HH_c^T$ (line 294), and not $HE^T + HE_{c}^{T}$. We will use equation 3 in Line 233 to explain these terms. $A$ is the log-probability matrix. It is the log_softmax of output logits. Under the cache-LM setting, output logits are the sum of the original token logits $HE^T$ (first term of equation 3) and the cache logits $HH_c^T$ (the second term of equation 3). Essentially $A=$ log_softmax$(HE^T + HH_c^T)$. Property 2 of Yang et al. 2018 showed that the log-probability matrix and logit matrix have similar ranks, and so we say $A$ is roughly equivalent to $HE^T + HE_{c}^{T}$. As for $E_c$, it is equal to $E+H_c$, and we use it to show how the static output embedding matrix ($E$) becomes a context-dependent tensor ($E_c$) (line 295-300).
>
> We will expand on this point in the final version, thanks.
>
>
>
>
>
> **Performance on recent LLM**
>
> >The baselines are limited to GPT2 and BART. It would be interesting to see the performance of more recent works of LLM such as LLaMA.
>
> Due to academic computational constraints, we were not able to use very large models (note that we also finetune these models and not just zero-shot), but now we also run the LLaMA2 7B model on the ambiguous template. Interestingly, we find that LLaMA2 achieves 0% accuracy for Acc@{2,5,10} when evaluated zero-shot. After fine-tuning, the model achieves 100% accuracy without any cache. This is kind of expected, as the task is fairly simple, and the model, compared to gpt2-large, is 10x larger, and the hidden size is 3.2x larger (1280-> 4096), and we discussed how a larger model can alleviate softmax bottleneck in Section 4.1 (Line 279-281).
>
> However, we still observe the two problems with LLaMA2. First, the problem of softmax bottleneck still exists, as the rank of its output log-probability matrix ($A$) is still upper-bounded by its hidden size of 4096. We find that its empirical rank is 3332. This means that it is still theoretically less expressive than highly context-dependent natural language because ideally its rank can be as large as the vocabulary size (see Section 4.1). Second, TRIME on top of LLaMA2 is still not able to make good use of the cache (Line 319-323), i.e., misalignment still exists. As shown in the table below, TRIME achieves 0% accuracy for Acc@{2,5,10} under the cache-only setting, which shows that the issue of misalignment is even more apparent for larger language models: Since the token logits perform well enough, the model does not learn to use the cache anymore. Nevertheless, as shown in the table, our training objective can enforce the use of the local cache and achieve 100% accuracy, which is consistent with our findings from smaller models.
>
> The presence of these two issues showcases that there is probably still room for improvement on LM’s context dependency and our method still outperforms TRIME in making good use of cache. Our method thus may lead to gains in more complex downstream tasks, e.g., faithfulness for abstractive summarization that are not fully resolved by large models like LLaMA2.
>
>
> We will include this in the final version given the extra page.
>
> Cache-only results for LLaMa2 7B-based models:
>
> | **Model** | **Acc@2** | **Acc@5** | **Acc@10** | **Acc@25** |
> |-----------|-------|-------|--------|--------|
> | LLaMA2-7B | 0 | 0 | 0 | **100** |
> | TRIME | 0 | 0 | 0 | **100** |
> | HistAlign | **100** | **100** | **100** | **100** |

---

### Meta-Review · Area_Chair_rtHH · 2023-09-20

**Recommendation:** 4

**Metareview:**

This paper innovatively integrates the cache language model (Cache-LM) approach with the Transformer architecture to address incoherence and hallucination in state-of-the-art LMs. Its key contributions encompass bridging the cache LM strategy with the softmax bottleneck of Transformers, introducing a novel contrastive learning objective to enhance the similarity between histories associated with the same output token, proposing a discriminative training criterion with a max-margin loss, and adapting Cache-LM for both Transformer decoder and encoder-decoder architectures.
The reviewers commend the paper for being well-motivated and presenting a clear rationale. They highlight the innovative nature and logical soundness of methods such as the contrastive learning objective. The paper's strength is further bolstered by comprehensive evaluations that demonstrate significant improvement across multiple tasks. Additionally, the versatility of the proposed technique is emphasized, noting its applicability to various architectures, including decoder-only and encoder-decoder. Finally, the paper's coherence, ease of understanding, and the inclusion of supplementary code are appreciated.
The consolidated concerns from reviewers primarily focus on the paper's baseline model choices, emphasizing its reliance on older models like GPT-2 and BART and suggesting the inclusion of newer models like LLaMA. Additionally, there's a strong call for a more pronounced distinction and deeper discussion regarding the differences between Cache-LM and pointer networks. Most reviewers also raised questions about the proposed techniques' effectiveness as language models become larger and more sophisticated. Furthermore, there are requests for clarification on certain technical aspects of the paper, especially around specific lines and equations. Lastly, concerns were highlighted about some assumptions in the paper, particularly regarding the definition of terms in the equations and the general perspective on token probabilities.

---

### Decision · Program_Chairs · 2023-10-07

**Decision:**

Accept-Main

**Comment:**

This paper innovatively integrates the cache language model (Cache-LM) approach with the Transformer architecture to address incoherence and hallucination in state-of-the-art LMs. Its key contributions encompass bridging the cache LM strategy with the softmax bottleneck of Transformers, introducing a novel contrastive learning objective to enhance the similarity between histories associated with the same output token, proposing a discriminative training criterion with a max-margin loss, and adapting Cache-LM for both Transformer decoder and encoder-decoder architectures.
The reviewers commend the paper for being well-motivated and presenting a clear rationale. They highlight the innovative nature and logical soundness of methods such as the contrastive learning objective. The paper's strength is further bolstered by comprehensive evaluations that demonstrate significant improvement across multiple tasks. Additionally, the versatility of the proposed technique is emphasized, noting its applicability to various architectures, including decoder-only and encoder-decoder. Finally, the paper's coherence, ease of understanding, and the inclusion of supplementary code are appreciated.
The consolidated concerns from reviewers primarily focus on the paper's baseline model choices, emphasizing its reliance on older models like GPT-2 and BART and suggesting the inclusion of newer models like LLaMA. Additionally, there's a strong call for a more pronounced distinction and deeper discussion regarding the differences between Cache-LM and pointer networks. Most reviewers also raised questions about the proposed techniques' effectiveness as language models become larger and more sophisticated. Furthermore, there are requests for clarification on certain technical aspects of the paper, especially around specific lines and equations. Lastly, concerns were highlighted about some assumptions in the paper, particularly regarding the definition of terms in the equations and the general perspective on token probabilities.